# Incidence of Acute Kidney Injury in Polytrauma Patients and Predictive Performance of TIMP2 × IGFBP7 Biomarkers for Early Identification of Acute Kidney Injury

**DOI:** 10.3390/diagnostics12102481

**Published:** 2022-10-13

**Authors:** Gianlorenzo Golino, Massimiliano Greco, Alessandro Rigobello, Vinicio Danzi, Massimo De Cal, Nicola Malchiorna, Monica Zannella, Paolo Navalesi, Rahul Costa-Pinto, Claudio Ronco, Silvia De Rosa

**Affiliations:** 1International Renal Research Institute of Vicenza, 36100 Vicenza, Italy; 2Department of Anesthesiology and Intensive Care, San Bortolo Hospital, 36100 Vicenza, Italy; 3Department of Medicine—DIMED, Section of Anesthesiology and Intensive Care Medicine, University of Padova, 35100 Padova, Italy; 4Department of Anesthesiology and Intensive Care, IRCCS Humanitas Research Hospital, 20100 Milan, Italy; 5Department of Biomedical Science, Humanitas University, 20100 Milan, Italy; 6Department of Nephrology, Dialysis and Transplantation and International Renal Research Institute of Vicenza, San Bortolo Hospital, 36100 Vicenza, Italy; 7Department of Intensive Care, Austin Hospital, 145 Studley Road, Melbourne, VIC 3084, Australia; 8Department of Critical Care, the University of Melbourne, Melbourne, VIC 3010, Australia

**Keywords:** IGFBP7, TIMP2, acute kidney injury, intensive care unit, outcomes, trauma

## Abstract

**Background:** Acute kidney injury (AKI) is a common cause of organ failure in trauma patients who survive their initial injuries. It is independently associated with increased morbidity and mortality and prolongs the length of hospital stays. The objectives of this study were to describe the incidence of early AKI and influence of risk factors in polytrauma patients and evaluate the predictive potential of TIMP2 × IGFBP7 biomarkers in this patient cohort. **Methods:** We conducted a retrospective cohort study of severely injured adult patients who were consecutively admitted to a multidisciplinary ICU from May 2017 to May 2019. Detailed patient data was retrieved from ICU medical records. Fluid balance, urinary output, and sCr values up to 72 h were collected. Urine samples for measuring TIMP2 × IGFBP7 concentrations were obtained and analyzed from ICU admission to 72 h. **Results:** Among the 153 patients eligible for analysis, 45 were included in the AKI, and 108 in the no AKI cohorts. The incidence of AKI within 72 h, based on KDIGO criteria, was 28.8%. There were no differences in ISS, type and mechanism of injury, heart rate, and systolic BP at admission between groups. Patients with early AKI were older (68 vs. 49 years, *p* < 0.001), obese (BMI 26.2 vs. 24.7, *p* < 0.048), and more likely to have previous cardiac disease (27% vs. 5.6%, *p* < 0.001). TIMP2 × IGFBP7 values on ICU admission were associated with subsequent AKI in patients without evidence of AKI at the time of ICU admission. They were also higher in the AKI cohort and significantly correlated with renal replacement therapy (RRT) and episodes of hypotension. Multivariable analysis confirmed age, previous cardiac disease, and NephroCheck as the variables mostly associated with AKI, with AUC 0.792. **Conclusions:** TIMP2 × IGFBP7 may help identify trauma patients with tubular damage that may evolve into a clinically manifested syndrome. Prospective studies of TIMP2 × IGFBP7, as a trigger for early AKI bundle care, are warranted.

## 1. Background

Acute kidney injury (AKI) is a common cause of organ failure in trauma patients who survive their initial injuries. It is independently associated with increased morbidity and mortality and prolongs the length of hospital stay [1,2,3,4]. The incidence of trauma-associated AKI in injured patients admitted to an intensive care unit (ICU) is approximately 20–24%, and one in 10 patients who develop AKI will require renal replacement therapy (RRT), corresponding to approximately 2% of the total trauma population [5,6]. A number of risk factors, such as hypotension, hypoperfusion, hypovolemia, inflammation, rhabdomyolysis, and nephrotoxic effects of therapies have been found to be associated with AKI [7,8]. Several prophylactic strategies have been tested, but the degree to which the development of trauma-associated AKI may be prevented remains unresolved [9]. The high incidence of AKI in trauma patients should prompt the early identification of those at risk of AKI to establish a resuscitation strategy that aims to prevent this complication. However, current evidence on trauma patients reflects the lack of uniform AKI definitions, based on KDIGO criteria, while the prediction of AKI based on serum creatinine or urinary output alone is unreliable.

There is also a new category of AKI to take into consideration: subclinical AKI, a syndrome with elevations of tubular damage biomarkers alone, which may evolve to clinical dysfunction [10]. Subclinical AKI can be identified with novel biomarkers, and in this category, cell-cycle arrest urinary biomarkers, such as insulin-like growth factor-binding protein 7 (IGFBP7) and tissue inhibitor of metalloproteinases-2 (TIMP2), have been shown to have a superior profile of accuracy and stability, even in patients with common comorbidities [11]. In September 2014, the United States Food and Drug Administration (FDA) approved the clinical use of the arithmetic products of IGFBP7 and TIMP2, known as the NephroCheck test (Astute Medical, San Diego, CA, USA) (united in (ng/mL)^2^/1000), to be used in ICU patients to predict the risk of developing moderate to severe AKI. The test is based on a sandwich immunoassay technique. A 0.3 cut-off has been established to achieve high sensitivity, while preserving acceptable specificity [12,13].

In our institution, we conduct an assessment of TIMP2 × IGFBP7 values, and if the AKI Risk score is positive (i.e., >0.3), we implement an AKI care bundle [14,15,16,17,18,19,20,21,22]. This bundle, defined as “a structured method of improving processes of care and patient outcomes”, consists of an e-alert for AKI, fluid balance and volume assessment, urine diagnostic tests, medication adjustment, avoidance of nephrotoxic agents (angiotensin-converting enzyme inhibitors, angiotensin receptor blockers, radiocontrast agents, non-steroidal anti-inflammatory drugs, amphotericin, vancomycin, aminoglycosides, cyclosporine, tacrolimus, and N-acetylcysteine), follow-up by a nephrologist, and escalation of therapy or palliative care, if necessary [22].

The objectives of this study were to describe the incidence of early AKI and influence of risk factors in polytrauma patients and evaluate the predictive potential of TIMP2 × IGFBP7 biomarkers in this patient cohort.

## 2. Methods

### 2.1. Study Design

We conducted a retrospective cohort study of severely injured adult patients consecutively admitted to a multidisciplinary ICU from May 2017 to May 2019. All adult patients (age ≥ 18 years) with a diagnosis of polytrauma were included. This was defined, in terms of their injury severity score (ISS), whereby patients with an ISS ≥ 16 were included in the study. We excluded patients with isolated traumatic brain injuries (TBI), burns, length of ICU stay < 24 h, death in ICU within 24 h, missing data on relevant outcomes, and previous hospitalization in another hospital. The study was approved by the local Ethics Committee of the San Bortolo Hospital (Vicenza), with protocol number 03/17, and conformed to the Declaration of Helsinki. Although this was a retrospective study, informed consent for patient data collection and use was routinely obtained from all patients at ICU admission, according to our local institution regulations and Italian laws.

### 2.2. Definitions

Baseline serum creatinine (sCr) level was defined as the most recent sCr within 3–6 months prior to admission. If this data was unavailable, the admission sCr was considered the baseline sCr [23]. Back-calculating baseline sCr was not considered, because in patients with suspected chronic kidney disease (CKD), it overestimates the incidence of AKI [24]. AKI was defined and staged according to the KDIGO 2012 consensus guidelines [25], from ICU admission to 72 h. AKI stage 1 was defined as an absolute increase in sCr of ≥0.3 mg/dL, a percentage increase in sCr of ≥50% (1.5-fold from baseline), or a reduction in urinary output (documented oliguria of <0.5 mL/kg per hour for more than six hours) [23]. AKI stage 2 was defined as a 2.0- to 2.9-fold increase in sCr from baseline, and AKI stage was defined as a 3- to 3.0-fold increase in sCr from baseline, or an absolute increase in sCr of ≥4.0 mg/dL or any AKI treated with RRT [17]. For each time point, sCr considered for AKI diagnosis was corrected for fluid balance, according to recent evidence [24].

Progression of AKI was defined as an increase in sCr ≥ 1.5 times baseline, which was known or assumed to have occurred within the preceding 7 days, whereas regression was defined as a decrease in sCr ≤ 1.5 times the AKI sCr level for the patient within 7 days. Renal recovery was defined as a return to within 50% above baseline sCr, as per the ADQI definition [26].

The sequential organ failure assessment (SOFA) score from ICU admission to 72 h was calculated using standard methods [12]. NephroCheck test, measuring urinary TIMP2 and IGFBP7, was performed. TIMP2 × IGFBP7 value > 0.3 (ng/mL)^2^/1000 was considered positive, while a value ≤0.3 was considered negative [15]. The final test output, labeled ‘AKI Risk’, is shown as a numeric score and was considered a continuous variable in this study.

Daily fluid balance was recorded from ICU admission to 72 h as the algebraic sum of fluid intake and output per day, not including insensible losses, while cumulative fluid balance (CFB) was calculated as the algebraic sum of daily fluid balance during the observational period. Cumulative fluid overload (FO) was calculated by dividing the CFB by the admission weight of each patient. The result was expressed as a percentage. We considered an FO between 5% and 9.99% as moderate and an FO ≥ 10% as severe, based on the published literature [27,28].

Injuries were classified according to the ISS by certified coders. The ISS describes injury severity on the basis of anatomical findings defined in the abbreviated injury scale (AIS). The severity of each individual injury was graded on a scale from 1 to 6 points, where 1 point describes minor injuries, and 6 points were given for untreatable, mostly lethal injuries. In order to calculate the ISS, each AIS score was assigned to one out of six different body regions [29]. The ISS was calculated as the sum of the squares of the highest AIS code (each ranging from 0 to 6) in each of the 3 most severely injured ISS body regions (6 body regions in total: head or neck, face, chest, abdominal or pelvic contents, extremities or pelvic girdle, external). The ISS is widely used in trauma outcomes research to dichotomize patients by injury severity. Traditionally, most studies define severely injured trauma patients by an ISS ≥ 16, which has been associated with a mortality risk of 10% in a study from 1987 [16,17,18,19,20,21,22,23,24,25,26,27,28,29,30].

### 2.3. Study Endpoints

The primary outcome was the assessment of AKI incidence, based on KDIGO criteria in ICU polytrauma patients within 72 h, as well as its related risk factors.

Secondary outcomes were AKI predictive performance of TIMP2 × IGFBP7 urinary biomarkers at ICU admission, need for RRT, ventilation-free days, hospital and ICU length of stay, development of persistent chronic kidney disease, renal recovery, and survival status.

### 2.4. Data Collection

Detailed data of trauma patients between May 2017 and May 2019 were retrieved from ICU medical records. This included patients’ demographics, anthropometry, comorbidities, ISS, SOFA score on admission, hemodynamic parameters, biochemical parameters, fluid balance, urinary output, and sCr values at baseline, 24 h, 48 h, and 72 h. Serum creatinine was measured using the enzymatic method with an automatic analyzer (Dimension Vista, Siemens Healthcare, Tarrytown, NY, USA). The concentration of biomarkers TIMP2 and IGFBP7 was analyzed with the Astute 140 m Platform (Ortho Clinical Diagnostics), using NephroCheck kits (Astute Medical). The Astute 140 m (Astute Medical) divided the concentrations of the two biomarkers by 1000 to report a single numerical value in units of (ng/mL)^2^/1000. Urine samples for measuring TIMP2 and IGFBP7 concentrations were obtained and analyzed at ICU admission through to 72 h. The NephroCheck kits were acquired by the central laboratory at market price. In order to develop a new diagnostic tool, the hospital allocated a special budget. Data on continuous RRT, renal recovery at ICU discharge and at 60 days follow-up, ICU discharge, and death were also recorded. Adverse events during the ICU stay were recorded.

### 2.5. Statistical Analyses

Sample size was calculated as 80, assuming α as 0.05 and β as 0.02 with 95% power. Values were expressed as frequencies and median interquartile ranges (IQR). Chi-square test was used to assess differences between categorical variables, while Mann–Whitney U-test was used to assess differences in continuous variables. Multivariable logistic regression was conducted, including two clinically important trauma variables by default (ISS and shock index). Other variables were selected using backward stepwise regression, according to the lowest Aikake information criterion (AIC). AIC depends on the number of independent variables included in the model and on the maximum likelihood estimation of each model. The best model chosen, according to AIC, is the one explaining the greatest amount of variation using the fewest possible independent variables. Strength of associations between continuous variables were assessed through Spearman correlation, calculating correlation coefficients for each combination. Results were represented as heat maps. All statistical analysis was performed using R software v 4.1.1. A *p* value < 0.05 was considered statistically significant at univariate analysis, and a *p* value < 0.1 was considered statistically significant at Wald test for multivariable analysis.

## 3. Results

### 3.1. Patients Characteristics

Flowchart of the study design is shown in Figure 1. During the study period, 206 patients underwent screening. Fifty patients were excluded: 23 with isolated TBI, 12 with hospitalization less than 24 h, 2 for death within 24 h, 4 for burns, 6 for admission to another hospital, and 3 for missing data. Among 153 patients eligible for analysis, 45 were included in the AKI, and 108 in the no AKI cohorts. A documented baseline creatinine was present in 42 patients, and admission creatinine was used as the reference creatinine in 111 patients. The baseline characteristics associated with AKI are reported in Table 1. There were no differences for ISS, type and mechanism of injury, Glasgow Coma Scale (GCS), heart rate, and systolic blood pressure at admission. Although not significant, the shock index was higher in the AKI cohort. In addition, 4.4% of these patients had previous renal disease. Patients with AKI within 72 h were older (68 vs. 42 years, *p* < 0.001) and overweight (BMI 26.2 vs. 24.7, *p* < 0.048). Forty-nine percent had a history of hypertension and nephrotoxic drugs use, 40% used antiplatelet and anticoagulation therapy, and 27% had previous cardiac disease (*p* < 0.001).

### 3.2. AKI Development and TIMP2 × IGFBP7

In the overall population, the median value of sCr at admission was 0.84 (0.70, 1.01) mg/dL. These values were, respectively, at 24 h, 48 h, and 72 h: 0.82 (0.70, 0.99), 0.81 (0.68, 1.07), and 0.75 (0.64, 0.92) mg/dL. Of the 153 patients, 45 (28.8%) developed AKI within 3 days in the ICU, of which 29 (18.9%) were categorized as AKI stage 1, and 16 (10.3%) were categorized as AKI stage 2–3; a total of 16.99% (n = 26) of patients had an AKI at ICU between ICU admission, and at 24 h, and 12.42% (n = 19) had an AKI between 24 and 48 h. In the overall population, urine output increased from 24 to 72 h. Differences were observed for urinary output between the AKI and no AKI cohorts at 24 h (*p* = 0.003) and 48 h (*p* = 0.005), associated with more furosemide use, as shown in Table 2. TIMP2 × IGFBP7 values were higher in AKI patients, compared with those who did not develop AKI (0.63 (0.13–2.19) vs. 0.23 (0.06–0.59), *p* < 0.001). AKI risk score at ICU admission was associated with subsequent AKI in patients without evidence of AKI at time of ICU admission, while creatinine levels were not associated with AKI incidence. Similarly, most cases of AKI had biological markers (78%), while 5% of cases had functional markers only (Appendix A). Of the patients with isolated elevated urinary biomarkers who received the AKI care bundle, only 8.5% developed stage 2–3 AKI, compared with 33% of patients with creatinine elevation. A total of 50% of patients had both increased creatinine and AKI risk score. Nineteen patients (23%) with urinary marker-driven interventions progressed to higher stages of AKI, while seven (8.5%) improved their AKI stage. In patients with AKI, there was no association between TIMP2 × IGFBP7 values and progression or regression of AKI (*p =* 0.47 for progression and *p =* 0.22 for regression of AKI) (Figure 2). Association between variables were reported in heat maps (Appendix A). There was a significant correlation between TIMP2 × IGFBP7 values at admission and RRT (*p =* 0.031), as well as between TIMP2 × IGFBP7 values at admission and episodes of hypotension (*p =* 0.049) (Figure 3).

### 3.3. Renal Outcomes

Our results showed that AKI patients had a significantly higher base excess (BE) at admission (*p =* 0.038), higher myoglobin levels at 48 h and 72 h (*p* < 0.001 and *p =* 0.023), more acidic urine pH at 48 h (*p =* 0.03), significant albumin use at admission, and a more positive fluid balance from 24 h to 72 h, compared to the no AKI cohort (*p* < 0.05). There was no significant difference in the percentage of FO at 48 h (*p =* 0.042). There was no difference between the two groups, concerning Hb values. In addition, patients with AKI had a significantly lower MAP at 72 h (*p =* 0.038), higher lactate level at admission and at 24 h (*p =* 0016 and *p =* 0.002), a worse trend of SOFA scores from admission to 24 h (*p* < 0.05), and no significant difference in vasoactive inotropic score (VIS), but a vasopressor period that was longer (1 (0,1) vs. 1 (0–2) days, *p* < 0.001) (Table 3 and Table 4). Renal replacement therapy was used in five patients (31% of severe AKI cases). Of note, one patient with CKD on regular hemodialysis, following a previous trauma, was transferred to the nephrology ward on day three. Regarding cardiorespiratory variables, AKI patients had a lower P/F ratio at 24 h (*p =* 0.001) and 48 h (*p =* 0.031), compared to no AKI patients, but there was no significant difference in PaO_2_ values and PEEP between the two groups. The duration of mechanical ventilation was different between the AKI and no AKI groups (3 vs. 4 days, *p* = 0.038). Complications and length of stay are reported in Table 4, Table 5 and Table 6. Thirty-seven (82.2%) AKI patients achieved renal recovery at ICU discharge, while 24 (53.3%) maintained recovery at 60 day follow-up.

In the multivariate regression analysis, age was significantly associated with AKI (OR 1.04, 95% CI 1.02–1.07 per year, *p =* 0.002), while the shock index and ISS were not statistically significant. Previous cardiac disease (OR 3.75, 95% CI 1.05–15, *p =* 0.048) was the only comorbidity significantly associated with AKI. The TIMP2 × IGFBP7 values reached borderline significance (for each AKI risk score point, OR 1.53 (0.99–2.40), *p =* 0.055) (Appendix A). As shown in the multivariate analysis, age (*p =* 0.002) and previous cardiac disease (*p =* 0.048) remained significantly associated with incidence of AKI. The TIMP2 × IGFBP7 values reached borderline significance, *p =* 0.055. The area under the curve for a multivariable prediction model was 0.792 (Appendix A). The probability of AKI, given the AKI risk admission values, is reported in Figure 4. There was no significant difference for ICU and hospital length of stay, nor ICU and hospital mortality. In Appendix A, the renal outcomes, according to the TIMP2 × IGFBP7 values at admission, are reported.

## 4. Discussion

In our cohort of adult polytrauma patients with an ISS ≥ 16, the incidence of early AKI within 72 h of ICU admission was almost 29%, and the need for renal replacement therapy was almost 15%. Over 80% of patients with an AKI achieved renal recovery by ICU discharge. Age, obesity, and previous cardiac disease were significantly associated with incidence of AKI, whereas the shock index, ISS, and TIMP2 × IGFBP7 values were not. The TIMP2 × IGFBP7 values on ICU admission were associated with subsequent AKI in patients without evidence of AKI at the time of ICU admission. They were also higher in the AKI cohort and significantly correlated with renal replacement therapy (RRT) and episodes of hypotension.

Since the development of AKI consensus guidelines, several studies have reported the incidence of AKI in trauma patients, based on different diagnostic criteria [8,9,10]. Haynes et al. [18] showed pooled incidence of AKI after major trauma of 20.4%. In addition, the authors found a trend toward increasing mortality with worsening AKI stage, consistent with other groups of critically ill patients. Søvik et al. reported an incidence of AKI of approximately 24% in trauma patients admitted to the ICU. Only 4% had severe AKI, and less than 2% of all trauma patients were treated with RRT [5].

The higher percentage of patients in our study with AKI, compared with other contemporary studies, may reflect a higher ISS inclusion criterion, although ISS was not independently associated with incidence of AKI. More likely, the correction of sCr for fluid balance and the use of the KDIGO criteria to define and classify AKI in our study allowed for the identification of more patients with AKI. Our use of a fluid balance-adjusted creatinine to stage AKI is the first available in the literature on trauma patients, and it is strongly recommended, since the recent studies have shown underestimation and misclassification of AKI if uncorrected creatinine is used [19,20,21,22]. The assessment of renal injury based on the KDIGO criteria must take all factors that can result in elevated creatinine into account, such as fluid balance, muscle mass, age, body weight, and race [23,24,25,26].

AKI after major trauma has a clearly defined time of insult, making it an attractive target for potential interventions, in order to prevent AKI [11]. In the AKI group, the TIMP2 × IGFBP7 values, assessed at admission, were significantly higher, without evidence of AKI. This is consistent with Hatton et al.’s study, which found that urinary TIMP2 × IGFBP7 measured on ICU admission accurately predicted 48-h AKI and was independently associated with AKI and dialysis requirement after trauma.

The use of TIMP2 × IGFBP7 biomarkers at admission could, therefore, be helpful in the trauma setting, where implementation of an AKI care bundle to prevent or interrupt the mitigation of kidney damage is possible. This may include, for example, an automated electronic alert, which prompts the physician to intervene by following a checklist of preventive measures. Kane-Gill et al. [31] reported the use of urinary TIMP2 × IGFBP7 as an AKI risk screening tool, which resulted in the differential application of various components of the KDIGO bundle for patient management for those with a positive test result. In surgical settings, goal-directed volume management in patients with positive TIMP2 × IGFBP7 results has shown an improvement in patient outcomes [32,33]. Similar bundles may be beneficial for the severely injured patient population, as highlighted by our study, where, of the patients with isolated elevated urinary biomarkers who received the AKI care bundle, only 8.5% developed stage 2–3 AKI.

NephroCheck (TIMP2 × IGFBP7) is performed routinely at ICU admission in our institution, with the implementation of an AKI care bundle, as required, and appears feasible. We, therefore, recommend the use of biomarkers of early renal damage in patients admitted to ICU with ISS ≥ 16 and the application of the AKI care bundle, if positive. Unfortunately, in our patients with AKI, there was no association between TIMP2 × IGFBP7 and the progression or regression of AKI, in contrast to the data presented in the literature. Prospective studies are, therefore, warranted to evaluate the AKI outcomes associated with this approach.

A recent modified Delphi panel report [34] agreed that patients with severe trauma are a target population for urinary TIMP2 × IGFBP7 testing, especially those who have hemodynamic instability. However, the relationship between hypotension and AKI in an ICU setting is not completely understood. Lehman et al. [35] found the severity and duration of hypotension as significant risk factors in AKI development in ICU patients. Based on our findings, a correlation between TIMP2 × IGFBP7 at admission and episodes of hypotension could address further studies on the effective “time window of opportunity” for correcting hypotension.

Cho et al. [36] showed urinary responsiveness of TIMP2 × IGFBP7 on the first day of AKI diagnosis to predict RRT demand, as well as functional recovery. In our study, over 80% of patients had renal recovery at ICU discharge. Kidney function seems to recover well in most trauma patients with AKI, but there is a lack of data on the risk of CKD and long-term mortality [5]. The evaluation of renal recovery across populations is challenging because there is no consensus definition as of yet [37]. Although the impact of biomarkers and AKI care bundle application could have a positive impact on ICU and hospital length of stay, development of CKD, renal recovery, and mortality [32], future research evaluating the clinical and cost-effectiveness of the test-guided implementation of protective care bundles is necessary.

We acknowledge the following limitations in our study. First, this was a retrospective study, and multiple unmeasured variables may have affected the outcomes. Our conclusions need to be validated by larger, prospective studies in the future. Second, we did not adjust the creatinine for surrogate markers of muscle mass, such as age, body weight, and race. Third, we did not define AKI as a condition when AKI stage 1 or greater was present 7 days after an AKI initiating event, and we excluded patients who died or were discharged from the hospital within the observation period. Fourth, we did not plot ROC curves predicting AKI. Fifth, our study was not designed to predict the necessity of RRT.

## 5. Conclusions

In our study, the incidence of AKI, based on KDIGO criteria, was 28.8% after polytrauma. A single urinary TIMP2 × IGFBP7 test can identify trauma patients early who are at risk of developing AKI, compared with the current methods based on sCr measurement. Future prospective studies should be conducted in this population to evaluate whether the use of urinary TIMP2 × IGFBP7, as a tool for AKI risk assessment, can lead to a lower incidence of AKI and reduction in mortality with the differential application of various components of the KDIGO care bundle.

## Figures and Tables

**Figure 1 diagnostics-12-02481-f001:**
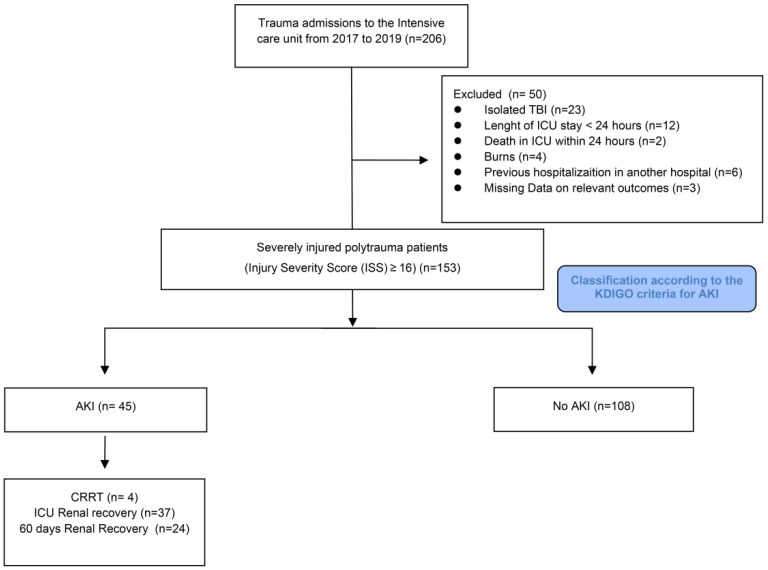
Study flowchart.

**Figure 2 diagnostics-12-02481-f002:**
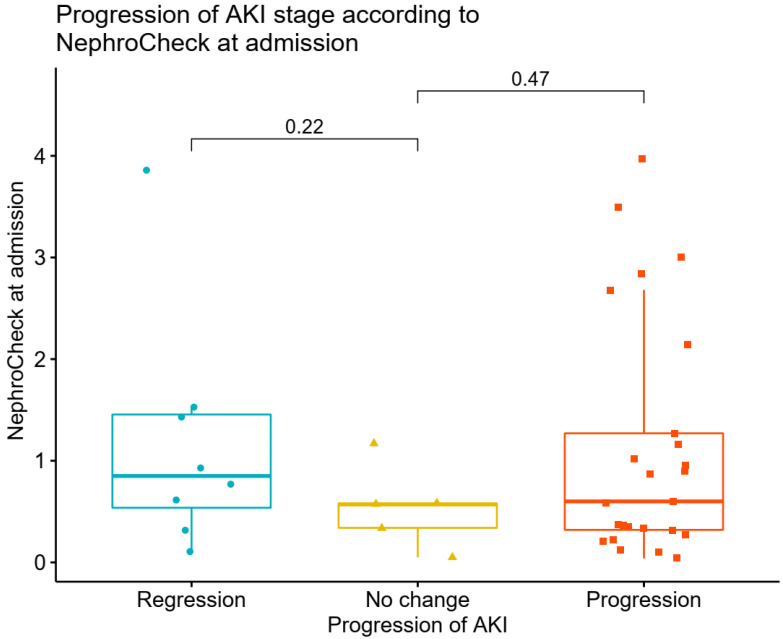
Association between NephroCheck at ICU admission and progression or regression of AKI.

**Figure 3 diagnostics-12-02481-f003:**
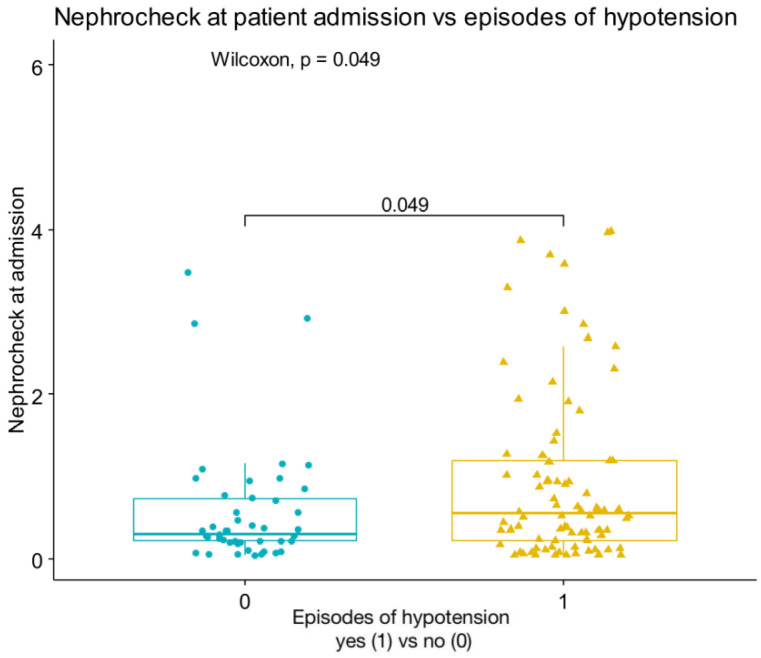
Correlation between NephroCheck at ICU admission and episodes of hypotension.

**Figure 4 diagnostics-12-02481-f004:**
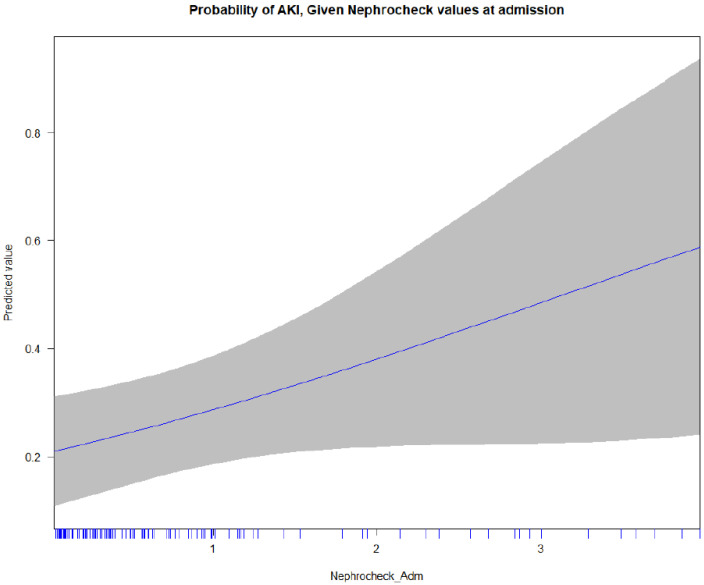
Probability of AKI given NephroCheck admission values.

**Table 1 diagnostics-12-02481-t001:** Baseline characteristics, comorbidities, medications, and mechanism of injury in the overall population and according to AKI status.

Characteristic	Entire Cohort N = 153 ^1^	AKI N = 45 ^1^	No AKI N = 108 ^1^	*p*-Value ^2^
**Patient history and chronic medications**				
Age	55 (36, 72)	68 (54, 79)	49 (31, 66)	**<0.001**
Gender				0.2
Female	26 (17%)	5 (11%)	21 (19%)	
Male	127 (83%)	40 (89%)	87 (81%)	
BMI	24.8 (22.9, 27.8)	26.2 (23.9, 28.4)	24.7 (22.9, 27.7)	**0.048**
Type 2 diabetes	14 (9.2%)	6 (13%)	8 (7.5%)	0.4
Hypertension	41 (27%)	22 (49%)	19 (18%)	**<0.001**
Antiplatelet and anticoagulation therapy	30 (20%)	18 (40%)	12 (11%)	**<0.001**
Previous cardiac disease	18 (12%)	12 (27%)	6 (5.6%)	**<0.001**
Previous renal disease	3 (2.0%)	2 (4.4%)	1 (0.9%)	0.2
COPD	1 (0.7%)	1 (2.2%)	0 (0%)	0.3
**Status on admission**				
GCS at admission	15.0 (12.0, 15.0)	15.0 (13.0,15.0)	15.0 (10.5, 15.0)	0.2
Systolic BP at admission	130 (109, 150)	124 (105, 148)	133 (110, 151)	0.2
Heart rate at admission	88 (74, 104)	84 (73, 104)	89 (76, 103)	0.7
Shock index	0.69 (0.55, 0.84)	0.70 (0.54, 1.00)	0.68 (0.55, 0.81)	0.5
ISS	17 (12, 24)	18 (13, 22)	17 (12, 24)	0.8
TRISS (based on ISS)	0.96 (0.89, 0.98)	0.96 (0.89, 0.98)	0.96 (0.89, 0.99)	0.2
TRISS (based on NISS)	0.94 (0.83, 0.98)	0.94 (0.86, 0.98)	0.94 (0.83, 0.98)	0.3
**Type of injury**				0.3
Blunt	140 (92%)	40 (89%)	100 (93%)	
Penetrating	12 (7.9%)	5 (11%)	7 (6.5%)	
**Mechanism of injury**				0.6
Bicycle hit	21 (14%)	4 (8.9%)	17 (16%)	
Fall < 6 m	28 (18%)	8 (18%)	20 (19%)	
Fall > 6 m	5 (3.3%)	2 (4.4%)	3 (2.8%)	
Gunshot wound	2 (1.3%)	2 (4.4%)	0 (0%)	
Machinery	4 (2.6%)	1 (2.2%)	3 (2.8%)	
Motor vehicle traffic accident	24 (16%)	8 (18%)	16 (15%)	
Motorcycle	31 (20%)	10 (22%)	21 (20%)	
Other	17 (11%)	4 (8.9%)	13 (12%)	
Pedestrian	17 (11%)	6 (13%)	11 (10%)	
Stab wound	3 (2.0%)	0 (0%)	3 (2.8%)	

^1^ Median (IQR); n (%); ^2^ Wilcoxon rank sum test; Pearson’s Chi-squared test; Fisher’s exact test. Abbreviations: BMI, body mass index; GCS, Glasgow coma scale; TRISS, trauma injury severity score; ISS, injury severity score; NISS, new injury severity score; COPD, chronic obstructive pulmonary disease.

**Table 2 diagnostics-12-02481-t002:** Fluid balance, urinary output, biomarkers, need for RRT, and inflammatory markers in the entire cohort and according to AKI.

Characteristic	Entire Cohort N = 153 ^1^	AKI N = 45 ^1^	No AKI N = 108 ^1^	*p*-Value ^2^
**sCr admission**	0.84 (0.70, 1.01)	1.12 (0.95, 1.34)	0.78 (0.66, 0.90)	**<0.001**
**sCr 24 h**	0.82 (0.70, 0.99)	1.16 (0.97, 1.50)	0.76 (0.63, 0.87)	**<0.001**
**sCr 48 h**	0.81 (0.68, 1.07)	1.18 (1.00, 1.39)	0.73 (0.60, 0.84)	**<0.001**
**sCr 72 h**	0.75 (0.64, 0.92)	0.98 (0.86, 1.31)	0.69 (0.57, 0.80)	**<0.001**
**TIMP2 × IGFBP7 admission**	0.40 (0.21, 0.96)	0.60 (0.32, 1.25)	0.36 (0.15, 0.84)	**0.016**
**TIMP2 × IGFBP7 24 h**	0.22 (0.09, 0.69)	0.53 (0.06, 0.76)	0.18 (0.09, 0.60)	0.7
**TIMP2 × IGFBP7 48 h**	0.22 (0.12, 0.46)	0.29 (0.12, 0.56)	0.22 (0.14, 0.30)	0.8
**TIMP2 × IGFBP7 72 h**	4.98 (2.58, 7.38)	9.78 (9.78, 9.78)	0.18 (0.18, 0.18)	>0.9
**Myoglobin admission**	974 (458, 1888)	1632 (984, 2091)	638 (364, 1316)	0.065
**Myoglobin 24 h**	1354 (630, 2705)	1827 (969, 2,994)	944 (482, 2026)	0.10
**Myoglobin 48 h**	697 (258, 1595)	3025 (2063, 3502)	409 (188, 844)	**<0.001**
**Myoglobin 72 h**	835 (442, 2236)	1655 (968, 4860)	553 (71, 777)	**0.023**
**Glucose admission**	150 (128, 179)	161 (143, 210)	146 (127, 175)	**0.023**
**Glucose 24 h**	135 (118, 160)	141 (121, 168)	134 (117, 156)	0.2
**Glucose 48 h**	129 (114, 152)	130 (118, 160)	127 (110, 147)	0.13
**Glucose 72 h**	129 (113, 150)	133 (120, 159)	126 (110, 146)	0.11
**PCT admission**	0.27 (0.07, 1.15)	0.25 (0.10, 1.16)	0.27 (0.07, 1.15)	0.8
**PCT 24 h**	0.87 (0.32, 3.05)	1.13 (0.60, 3.33)	0.56 (0.30, 2.99)	0.8
**PCT 48 h**	0.9 (0.5, 3.9)	2.3 (0.8, 5.0)	0.8 (0.4, 2.3)	0.12
**PCT 72 h**	0.7 (0.3, 2.1)	1.5 (0.6, 4.4)	0.6 (0.2, 1.3)	0.13
**Crystalloids use 24 h**	152 (97%)	45 (100%)	107 (99%)	>0.9
**Crystalloids use 48 h**	113 (72%)	35 (78%)	78 (72%)	0.5
**Crystalloids use 72 h**	86 (55%)	26 (58%)	60 (56%)	0.8
**Colloids use 24 h**	59 (39%)	23 (51%)	36 (34%)	**0.044**
**Colloids use 48 h**	32 (28%)	12 (34%)	20 (26%)	0.3
**Colloids use 72 h**	16 (19%)	7 (27%)	9 (15%)	0.2
**FB 24 h**	1430 (457, 2636)	1803 (782, 3725)	1211 (269, 2289)	**0.031**
**FB 48 h**	850 (-16, 1938)	1580 (459, 2953)	648 (-66, 1557)	**0.017**
**FB 72 h**	860 (102, 1671)	1091 (524, 2315)	574 (-30, 1413)	**0.045**
**% FO 24 h**	18 (5, 35)	25 (9, 47)	17 (3, 33)	0.068
**% FO 48 h**	11 (0, 25)	22 (6, 27)	8 (-1, 23)	**0.042**
**% FO 72 h**	11 (1, 20)	16 (5, 24)	8 (0, 17)	0.14
**UO 24 h**	1855 (1289, 2575)	1402 (1160, 2050)	1945 (1525, 2800)	**0.003**
**UO 48 h**	2300 (1725, 3000)	1845 (1410, 2455)	2450 (1840, 3085)	**0.005**
**UO 72 h**	2570 (1888, 3140)	2120 (1702, 3530)	2590 (2035, 3012)	0.6
**Furosemide 24 h**	42 (27%)	19 (42%)	23 (21%)	**0.009**
**Furosemide 48 h**	41 (26%)	19 (42%)	22 (20%)	**0.007**
**Furosemide 72 h**	22 (14%)	7 (16%)	15 (14%)	0.9
**Urine pH admission**	5.25 (5.00, 6.75)	5.00 (5.00, 5.00)	5.50 (5.00, 7.00)	0.4
**Urine pH 24 h**	5.75 (5.00, 7.00)	5.25 (5.00, 6.50)	6.00 (5.50, 7.50)	0.058
**Urine pH 48 h**	7.50 (5.38, 8.50)	5.00 (5.00, 7.25)	7.50 (5.50, 8.50)	**0.003**
**Urine pH 72 h**	6.00 (5.50, 7.50)	7.50 (5.75, 8.25)	5.50 (5.25, 7.00)	0.2
**RRT 24 h**	1 (0.7%)	1 (2.2%)	0 (0%)	0.3
**RRT 48 h**	4 (3.5%)	4 (11%)	0 (0%)	**0.008**
**RRT 72 h**	4 (4.7%)	4 (15%)	0 (0%)	**0.007**

^1^ Median (IQR); n (%). ^2^ Wilcoxon rank sum test; Wilcoxon rank sum exact test. Abbreviations: FB, fluid balance; FO, fluid overload; PCT, procalcitonin; sCr, serum creatinine; RRT, renal replacement therapy; UO, urinary output.

**Table 3 diagnostics-12-02481-t003:** ICU admission scores for organ failure, hemodynamic data, and respiratory variables in the entire cohort and according to AKI.

Characteristics	Entire Cohort N = 153 ^1^	AKI N = 45 ^1^	No AKI N = 108 ^1^	*p*-Value ^2^
**VIS admission**	0.0 (0.0, 0.0)	0.0 (0.0, 0.0)	0.0 (0.0, 0.0)	0.4
**VIS 24 h**	0 (0, 5)	0 (0, 5)	0 (0, 5)	0.4
**VIS 48 h**	0 (0, 8)	0 (0, 10)	0 (0, 6)	0.2
**VIS 72 h**	0 (0, 7)	0 (0, 12)	0 (0, 7)	0.3
**MAP admission**	84 (69, 93)	79 (65, 88)	84 (72, 94)	0.066
**MAP 24 h**	78 (72, 89)	77 (71, 88)	80 (73, 90)	0.2
**MAP 48 h**	82 (73, 90)	81 (72, 89)	82 (74, 90)	0.4
**MAP 72 h**	83 (73, 91)	77 (72, 85)	86 (73, 91)	**0.038**
**Lactate admission**	1.70 (1.20, 3.00)	2.25 (1.40, 4.12)	1.70 (1.20, 2.50)	**0.016**
**Lactate 24 h**	1.40 (1.10, 2.00)	1.70 (1.20, 2.40)	1.30 (1.00, 1.90)	**0.002**
**Lactate 48 h**	1.10 (0.90, 1.50)	1.40 (0.90, 1.90)	1.00 (0.90, 1.30)	0.079
**Lactate 72 h**	1.00 (0.80, 1.20)	1.05 (0.83, 1.45)	0.90 (0.80, 1.20)	0.2
**SOFA score admission**	3.00 (2.00, 5.00)	4.00 (2.00, 6.25)	2.00 (1.00, 5.00)	**0.002**
**SOFA score 24 h**	5.0 (2.0, 9.0)	6.5 (3.8, 11.0)	4.0 (2.0, 8.0)	**0.001**
**SOFA score 48 h**	7.0 (3.0, 10.0)	9.0 (6.0, 12.0)	5.5 (2.0, 9.0)	**<0.001**
**SOFA score 72 h**	8.0 (5.0, 11.0)	10.0 (7.0, 11.0)	7.0 (4.0, 9.0)	**<0.001**
**Arterial BE admission**	-1.9 (-4.2, 0.3)	-2.7 (-5.8, -0.2)	-1.6 (-3.4, 0.5)	**0.041**
**Arterial BE 24 h**	2.30 (0.00, 3.80)	1.30 (-0.77, 3.92)	2.40 (0.30, 3.70)	0.4
**Arterial BE 48 h**	3.60 (1.90, 5.00)	3.30 (1.02, 5.20)	3.80 (2.20, 4.95)	0.4
**Arterial BE 72 h**	3.50 (1.90, 5.30)	3.95 (1.35, 5.97)	3.50 (2.10, 4.70)	0.6
**Arterial PaO_2_ admission**	110 (86, 170)	104 (86, 143)	119 (87, 174)	0.3
**Arterial PaO_2_ 24 h**	99 (81, 134)	91 (79, 109)	105 (81, 135)	0.081
**Arterial PaO_2_ 48 h**	102 (80, 130)	102 (85, 118)	103 (80, 138)	0.4
**Arterial PaO_2_ 72 h**	108 (89, 132)	105 (91, 123)	113 (88, 132)	0.7
**P/F ratio admission**	350 (265, 440)	340 (226, 422)	356 (274, 453)	0.12
**P/F ratio 24 h**	314 (242, 390)	268 (217, 324)	340 (250, 414)	**0.001**
**P/F ratio 48 h**	290 (227, 367)	250 (208, 320)	311 (238, 371)	**0.031**
**P/F ratio 72 h**	287 (211, 330)	262 (200, 303)	291 (224, 335)	0.10
**PEEP admission**	7.00 (6.00, 8.00)	6.00 (5.00, 8.00)	8.00 (6.00, 8.00)	0.12
**PEEP 24 h**	7.00 (6.00, 8.00)	8.00 (6.00, 8.00)	7.00 (6.00, 8.00)	0.7
**PEEP 48 h**	8.00 (6.00, 9.00)	8.00 (7.00, 10.00)	8.00 (6.00, 8.50)	0.2
**PEEP 72 h**	8.00 (7.00, 10.00)	8.00 (7.00, 10.00)	8.00 (7.75, 10.00)	>0.9

^1^ Median (IQR); n (%); ^2^ Wilcoxon rank sum test; Wilcoxon rank sum exact test. Abbreviations: BE, base excess; SOFA, sequential organ failure score; VIS, vasoactive inotropic score; MAP, mean arterial pressure; PEEP, positive end-expiratory pressure.

**Table 4 diagnostics-12-02481-t004:** Patient outcomes in the overall population and according to AKI.

Characteristics	Entire Cohort N = 153 ^1^	AKI N = 45 ^1^	No AKI N = 108 ^1^	*p*-Value ^2^
**Mechanical ventilation days**	1 (0, 8)	1 (0, 9)	1 (0, 8)	0.6
**Vasopressor use days**	1 (0, 2)	1 (0, 1)	1 (0, 2)	**<0.001**
**Length of ICU stay**	3 (1, 10)	6 (2, 10)	3 (1, 10)	0.6
**Length of hospital stay**	13 (7, 25)	11 (7, 27)	15 (7, 25)	>0.9
**ICU mortality**	7 (4.5)	2 (4.4%)	5 (4.5%)	>0.9
**Hospital mortality**	10 (6.4%)	5 (11%)	5 (4.5%)	0.2

^1^ Median (IQR); ^2^ Kruskal–Wallis rank sum test. Abbreviations: ICU: intensive care unit.

**Table 5 diagnostics-12-02481-t005:** ICU stay variables, according to severity of AKI.

	Total	By AKI Severity			
	N = 153 ^1^	None, N = 108 ^1^	Stage 1, N = 29 ^1^	Stage 2–3, N = 16 ^1^	*p*-Value ^2^
**Days free of MV**	1.00 (0.00, 2.00)	1.00 (0.00, 2.00)	1.00 (1.00, 2.00)	1.00 (0.00, 3.25)	>0.9
**Days on ventilator**	1 (0, 8)	1 (0, 8)	1 (0, 6)	5 (1, 9)	0.5
**LOS**	13 (7, 25)	15 (7, 25)	12 (10, 29)	9 (6, 22)	0.7
**ICU stay**	3 (1, 10)	3 (1, 10)	2 (1, 10)	6 (3, 12)	0.3

^1^ Median (IQR); ^2^ Kruskal–Wallis rank sum test. Abbreviations: ICU, intenisve care unit; MV, mechanical ventilation; LOS, length of hospital stay.

**Table 6 diagnostics-12-02481-t006:** Adverse events during ICU stay in the overall population and according to development of AKI.

Characteristics	Entire Cohort N = 153 ^1^	AKI N = 45 ^1^	No AKI N = 108 ^1^	*p*-Value ^2^
**Episode of hypotension**	98 (64%)	35 (78%)	63 (59%)	**0.026**
**RBC transfusion**	80 (53%)	30 (67%)	50 (47%)	**0.025**
**Plasma transfusion**	38 (25%)	19 (42%)	19 (18%)	**0.001**
**Platelets transfusion**	20 (13%)	11 (24%)	9 (8.4%)	**0.008**
**Sepsis**	32 (21%)	9 (20%)	23 (21%)	0.8
**Multi-organ failure**	4 (2.6%)	0 (0%)	4 (3.7%)	0.3
**Coagulopathy**	28 (18%)	11 (24%)	17 (16%)	0.2
**Liver failure**	2 (1.3%)	1 (2.2%)	1 (0.9%)	0.5

^1^ n (%); ^2^ Chi-square test. Abbreviations: RBC, red blood cell transfusion.

## Data Availability

All data and related metadata underlying the future findings will be deposited in the Department of Anesthesia and Intensive Care, San Bortolo Hospital, Vicenza, Italy.

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
