# Peer review of "Incidence of Acute Kidney Injury in Polytrauma Patients and Predictive Performance of TIMP2 × IGFBP7 Biomarkers for Early Identification of Acute Kidney Injury"

_diagnostics, 2022, doi:10.3390/diagnostics12102481_

Round 1

Reviewer 1 Report (Previous Reviewer 3)

The authors have addressed most of the concerns. I have a few questions

- Regression of AKI- does this mean that Cr dropped below baseline or below the AKI Cr level for the patient. Baseline Cr generally means the stable Cr level which is generally normal for those without CKD. 

- From the figure it seems that nephrocheck detects those with no AKI also. This should be added to the discussion. 

I do not have any further comments. 

Author Response

Dear Reviewer,

We greatly appreciated the opportunity to revise our manuscript “Incidence of acute kidney injury in trauma patients and predictive performance of [TIMP-2] × [IGFBP7] biomarkers for early identification of acute kidney injury” (Diagnostics-1821684), submitted to the journal Diagnostics. We have made substantial revisions to address the comments of the reviewers, to whom we are grateful.

The authors have addressed most of the concerns. I have a few questions. Regression of AKI – does this mean that Cr dropped below baseline or below the AKI Cr level for the patient. Baseline Cr generally means the stable Cr level which is generally normal for those without CKD.

Response: We thank the reviewer for this question. The regression of AKI was the sCr decrease below the AKI sCr level for the patient and not the baseline sCr level. We have now stated this more clearly in the manuscript (pg 3, ln 118-120).

From the figure it seems that Nephrocheck detects those with no AKI also. This should be added to the discussion.

Response: We thank the reviewer for this comment. We have now emphasised this important finding in our rewritten discussion.

Reviewer 2 Report (New Reviewer)

Thank you for the opportunity to review the manuscript entitled “Incidence of Acute Kidney Injury and Predictive performance of TIMP2 and IGFBP7 biomarkers in trauma patients for early identification of acute kidney injury”.

I have several important comments and concerns about the manuscript that need to be explained and/or corrected.

- I would prefer to use "Nephrocheck" or "[TIMP-2] × [IGFBP7]" in the title of the article, since the article does not evaluate TIMP-2 and IGFBP7 biomarkers separately at all.

- In the Abstract, there is written, that the study was retrospective. But at the same place there is written, that: „TIMP2 and IGFBP7 were obtained immediately following enrollment.“ This is confusing. Similarly, in the methods, there is “retrospective” and “informed consent was obtained”. The authors need to explain more precisely what kind of study it was. If it was really retrospective, then NephroCheck was done in each patient as a local standard of care and data were later retrospectively evaluated for the needs of this study. Thus, the patients were not "enrolled" and could not sign consent to participate in the study in advance!

- There is some duplicate information in methods. E.g. there is no need to indicate in the inclusion criteria that only patients over 18 years of age were included and at the same time in the exclusion criteria that patients under 18 years of age were not included. It sounds a bit silly. Similarly: “ length of ICU stay less than 24 hours” vs. “death in ICU within 24 hours”.

- On the contrary, the exclusion criteria do not include burns here, it is only in study flowchart.

- In the methods, please state how "polytrauma" was defined according to the ISS.

- The objectives of the study are also confusedly described. The authors are most likely mixing the "outcome" of the patients and the aims of the study. In my opinion, the outcomes in this study will be: primary outcome - AKI incidence, secondary outcome – the need for RRT, ICU stay, time to renal recovery, development of persistent chronic kidney disease, mortality etc. And the aims of the study would be: to verify the predictive potential of the NephroCheck method, describe the influence of risk factors on the development of AKI, etc.

- Table 1 could be clearer. Have the authors divide the table into these sections in this order: Patient history and chronic medication, Status on admission, Type of injury, Mechanism of injury.

-
Why is the median (IQR) used? In methods it is indicated “mean +/-SD”.

- Time data/context is missing in the manuscript. How long patients were followed up to register AKI development, adverse events etc. ? For example, it is curious that in the group of patients who had sepsis as a complication AKI is less common... Can the authors explain this? What does AKI progression/regression mean -over how long of follow-up? What does the AKI risk score mean?

- Figure 1 shows that even in the "No AKI" patient group, "renal recovery" was evaluated. This sounds like nonsense. How can kidneys that do not fail recover? How did the authors define "Renal recovery" and how did they define it in the "No AKI" group?

- Why is there somewhere in the pictures "AKI risk score" and somewhere "NephroCheck"? Unite it.

-       Why VIS (VIS, vasoactive–inotropic score) was calculated? How often Inotropic support such as Dopamin, dobutamin, Milrinon etc. was used in trauma patients ?? Wouldn't it be better to simply use: noradrenaline dose/kg/min.

-       How was fluid overload assessed/defined?

- All tables and figures should have a much more detailed legend.

- The table shows that 4 patients needed RRT, but in biograph no. 4 only 3 patients are listed..?

- Also, only 4 patients with RRT is too few to present the data as significant. This limitation must be stated. Personally, I would exclude Figure 4.

- Which nephrotoxic drugs were counted? 

- I consider the discussion uninteresting. It only recapitulates the data obtained. The authors should focus more on what the study brings new to the clinical management of trauma patients. So, based on their data, do the authors recommend routine use of NephroCheck in polytrauma patients? What is the cost/benefit?

- The information about the fact that the authors'workplace routinely uses NephroCheck to identify patients at high-risk for AKI with subsequent use of clinical bundles, I recommend to mention immediately in the introduction or methods. It will enable the reader to better understand how the entire study was conducted.

- Its a pity! In this setting it would be great to conduct a prospective randomized study, whether NephroCheck+Bundle will lead to a lower incidence of AKI, or mortality.

-
Previous articles on NephroCheck+trauma, or TIMP2+trauma are completely missing from the references and discussion...

* In conclusion: I see the main concern about this manuscript in the insufficient novelties of this study. Overall, the manuscript seems very chaotic, unclear, disorganized. It is not clear what the main aim of the study was, whether to test NephroCheck or to evaluate the incidence of AKI and risk factors in trauma patients in general. Anyway, both have already been published many times.

Author Response

Dear reviewer, 

We greatly appreciated the opportunity to revise our manuscript “Incidence of acute kidney injury in trauma patients and predictive performance of [TIMP-2] × [IGFBP7] biomarkers for early identification of acute kidney injury” (Diagnostics-1821684), submitted to the journal Diagnostics. We have made substantial revisions to address the comments of the reviewers, to whom we are grateful.

I would prefer to use "Nephrocheck" or "[TIMP-2] × [IGFBP7]" in the title of the article, since the article does not evaluate TIMP-2 and IGFBP7 biomarkers separately at all.

Response: We thank the reviewer for this suggestion. We have changed the title accordingly.

In the Abstract, there is written, that the study was retrospective. But at the same place there is written, that: “TIMP2 and IGFBP7 were obtained immediately following enrollment.” This is confusing. Similarly, in the methods, there is “retrospective” and “informed consent was obtained”. The authors need to explain more precisely what kind of study it was. If it was really retrospective, then NephroCheck was done in each patient as a local standard of care and data were later retrospectively evaluated for the needs of this study. Thus, the patients were not "enrolled" and could not sign consent to participate in the study in advance!

Response: We thank the reviewer for this comment. This is a retrospective study but according to Italian laws for General Data Protection Regulation (GDPR), there is a need to collect informed consent for inclusion in any study even if it is for future retrospective work. We have removed the term "enrolled", as suggested, and replaced it with "included" (pg 3, ln 96-97). We have also added the following sentence – “Although this was a retrospective study, informed consent for patient data collection and use is routinely obtained from all patients at ICU admission according to our local institution regulations and Italian laws” (pg 3, ln 102-104).

There is some duplicate information in methods. E.g. there is no need to indicate in the inclusion criteria that only patients over 18 years of age were included and at the same time in the exclusion criteria that patients under 18 years of age were not included. It sounds a bit silly. Similarly: “length of ICU stay less than 24 hours” vs. “death in ICU within 24 hours”.

Response: We thank the reviewer for these suggestions. We have made suitable changes to avoid duplication of information in the inclusion and exclusion criteria (pg 3, ln 96-99). However, we have not combined “length of the ICU stay” and “death within 24 hours”, as we believe these to be two separate entities. Some patients in our ICU were discharged to a ward within 24 hours.

On the contrary, the exclusion criteria do not include burns here, it is only in study flowchart.

Response: Burns has now been added to the exclusion criteria (pg 3, ln 98).

In the methods, please state how "polytrauma" was defined according to the ISS.

Response: We thank the reviewer for this suggestion. A definition for polytrauma has now been included as follows: “This was defined in terms of their Injury Severity Score (ISS) whereby patients with an ISS ≥16 were included in the study” (pg 3, ln 96-97). We have also retained the definition of the ISS and how it is calculated to assist the reader in understanding our patient cohort characteristics (pg 4, ln 135-141).

The objectives of the study are also confusedly described. The authors are most likely mixing the "outcome" of the patients and the aims of the study. In my opinion, the outcomes in this study will be: primary outcome - AKI incidence, secondary outcome – the need for RRT, ICU stay, time to renal recovery, development of persistent chronic kidney disease, mortality etc. And the aims of the study would be: to verify the predictive potential of the NephroCheck method, describe the influence of risk factors on the development of AKI, etc.

Response: We thank the reviewer for these suggestions. We have more clearly stated our objectives, primary and secondary outcomes. Our objectives were as follows: 1) describe the incidence of acute kidney injury and influence of risk factors in polytrauma patients, and 2) evaluate the predictive potential of [TIMP2] x [IGFBP7] biomarkers in this patient cohort (pg 3, ln 89-91). Our primary outcome was “the assessment of AKI incidence, based on KDIGO criteria, in ICU polytrauma patients within 72 hours, and its related risk factors” (pg 4, ln 144-145). Our secondary outcomes were “AKI predictive performance of [TIMP2] × [IGFBP7] urinary biomarkers at ICU admission, need for RRT, ventilation-free days, hospital and ICU length of stay, development of persistent chronic kidney disease, renal recovery, and survival status” (pg 4, ln 146-149).

Table 1 could be clearer. Have the authors divide the table into these sections in this order: Patient history and chronic medication, Status on admission, Type of injury, Mechanism of injury.

Response: We thank the reviewer for this suggestion. We made changes accordingly (pg 6-7).

Why is the median (IQR) used? In methods it is indicated “mean +/-SD”.

Response: We apologise for this error and have made the appropriate correction (pg 4, ln 168).

Time data/context is missing in the manuscript. How long patients were followed up to register AKI development, adverse events etc.? For example, it is curious that in the group of patients who had sepsis as a complication AKI is less common. Can the authors explain this? What does AKI progression/regression mean – over how long of follow-up? What does the AKI risk score mean?

Response: We thank the reviewer for these important questions. We have added the necessary data collection timeframes (i.e. AKI within 72 hours; Adverse events during ICU stay). Sepsis is unlikely to occur in the first 72 hours of ICU polytrauma admission and this may reflect why there is no difference between the two groups in our study. 1 in 5 patients in both groups developed sepsis during their ICU stay. Furthermore, release of TIMP2 and IGFBP7 have been found to be the best predictors of the development of septic AKI and the application of a timely AKI care bundle may have mitigated sepsis related damage and progression to AKI.

AKI progression and regression have now been defined (pg 3, ln 118-120). The follow-up period was 7 days. The AKI risk score has also now been defined (pg 4, ln 126-127).

Figure 1 shows that even in the "No AKI" patient group, "renal recovery" was evaluated. This sounds like nonsense. How can kidneys that do not fail recover? How did the authors define "Renal recovery" and how did they define it in the "No AKI" group?

Response: We thank the reviewer for this observation and apologise for this oversight. Renal recovery was defined as a return to within 50% above baseline sCr, as per the ADQI definition, and this has now been added to the manuscript (pg 3, ln 121). Clearly, patients in the no-AKI group would not have renal recovery and this has now been removed from the manuscript.

Why is there somewhere in the pictures "AKI risk score" and somewhere "NephroCheck"? Unite it.

Response: Thanks for the comment. This has now been standardized to ‘Nephrocheck’.  

Why VIS (VIS, vasoactive–inotropic score) was calculated? How often Inotropic support such as Dopamin, dobutamin, Milrinon etc. was used in trauma patients ?? Wouldn't it be better to simply use: noradrenaline dose/kg/min.

Response: We thank the reviewer for this suggestion, however, we do not fully agree. The patient cohort in this study were critically ill with major polytrauma (Injury Severity Score ≥16). Severe trauma may be associated with significant hemorrhagic shock, impaired organ perfusion and potential trauma-related cardiac complications either from direct penetrating or blunt trauma or following severe extracardiac injuries (crush syndrome, septic complications). For this reason, we decided to utilise the vasoactive-inotropic score rather than simply noradrenaline dose, as we do occasionally use dopamine, dobutamine or adrenaline for these indications in our trauma cohort.

How was fluid overload assessed/defined?

Response: The definitions and assessment of fluid balance and fluid overload have now been included in the manuscript (pg 3-4, ln 128-132).

All tables and figures should have a much more detailed legend.

Response: We have now improved the legend in each table.

The table shows that 4 patients needed RRT, but in biograph no. 4 only 3 patients are listed?

Response: In Figure 4, only 3 patients were listed as one RRT patients did not have Nephrocheck values available. However, we have now removed figure 4, as suggested by the reviewer in the next comment, and this apparent inconsistency is therefore resolved.

Also, only 4 patients with RRT is too few to present the data as significant. This limitation must be stated. Personally, I would exclude Figure 4.

Response: We thank the reviewer for this suggestion, and have removed Figure 4.

Which nephrotoxic drugs were counted?

Response: The nephrotoxic drugs that were considered were angiotensin-converting enzyme inhibitors, angiotensin receptor blockers, radiocontrast agents, non-steroidal anti-inflammatory drugs, amphotericin, vancomycin, aminogly-cosides, cyclosporine, tacrolimus and N-acetylcysteine. We have now included these in the manuscript (pg 3, ln 84-87).

I consider the discussion uninteresting. It only recapitulates the data obtained. The authors should focus more on what the study brings new to the clinical management of trauma patients. So, based on their data, do the authors recommend routine use of NephroCheck in polytrauma patients? What is the cost/benefit?

Response: We thank the reviewer for this comment. We have now rewritten the discussion in full to focus on key findings, comparisons with previous studies and implications for practice. We have discussed the recommendation of routine use of Nephrocheck in polytrauma patients and also the need for future research evaluating the clinical utility and cost-effectiveness of its use.

The information about the fact that the authors' workplace routinely uses NephroCheck to identify patients at high-risk for AKI with subsequent use of clinical bundles, I recommend to mention immediately in the introduction or methods. It will enable the reader to better understand how the entire study was conducted.

Response: We thank the reviewer for this important suggestion. We have now included this information in our introduction to provide immediate context for our study (pg 3, ln 80-81).

It’s a pity! In this setting it would be great to conduct a prospective randomized study, whether NephroCheck+Bundle will lead to a lower incidence of AKI, or mortality.

Response: We fully agree with this comment and have added as such in our discussion and conclusion.

Previous articles on NephroCheck+trauma, or TIMP2+trauma are completely missing from the references and discussion...

Response: We have now added pertinent references and included discussion on Hatten et al.’s study as well as a recent modified Delphi panel report. There is no published literature on a Nephrocheck-guided implementation of AKI care bundles in the trauma setting, and hence we believe our study is unique.

In conclusion: I see the main concern about this manuscript in the insufficient novelties of this study. Overall, the manuscript seems very chaotic, unclear, disorganized. It is not clear what the main aim of the study was, whether to test NephroCheck or to evaluate the incidence of AKI and risk factors in trauma patients in general. Anyway, both have already been published many times.

Response: We thank the reviewer for their suggestions which have helped drastically improve our manuscript. We believe our manuscript now more clearly outlines our objectives, primary and secondary outcomes. Our discussion is now more focused and discusses key findings, comparisons with prior literature and implications for clinical practice. We have also completed a substantial English language edit to make the manuscript clearer and more organised. We disagree that our study findings have been published many times previously as, to our knowledge, this is the first study looking at a major polytrauma cohort (ISS ≥16) and the practical use of Nephrocheck with an AKI care bundle. We believe that this is a unique aspect to this study and adds to the literature on the use of renal biomarkers and AKI bundles.

Round 2

Reviewer 2 Report (New Reviewer)

The authors answered all the comments more or less sufficiently. The quality of the manuscript has clearly increased.

I just wanted to ask one more thing. Why do the authors state in the limitations: "Fourth, we did not plot ROC curves predicting AKI."?? Why don't they just do a ROC analysis and present the image and the AUC value? It would certainly be interesting and informative. in such case, I would also add AUC to the Manuscript abstract and conclusions.

Author Response

Reviewer #2 (Round 2):

The authors answered all the comments more or less sufficiently. The quality of the manuscript has clearly increased.

I just wanted to ask one more thing. Why do the authors state in the limitations: "Fourth, we did not plot ROC curves predicting AKI."?? Why don't they just do a ROC analysis and present the image and the AUC value? It would certainly be interesting and informative. in such case, I would also add AUC to the Manuscript abstract and conclusions

Response:  We thank the reviewer for the suggestion. We now add a ROC analysis as supplemental figure 2 and report Area Under the Curve as AUC in manuscript and in the abstract. We have added the ROC curve as supplemental figure 4 (pg 14, In 287-288).  

This manuscript is a resubmission of an earlier submission. The following is a list of the peer review reports and author responses from that submission.

Round 1

Reviewer 1 Report

This study evaluates the nephrocheck parameters in trauma patients. Unfortunately, the English writing is of poor quality. There are sentences when the reader needs to guess what the authors mean. One example is this: “In our study, two of AKI patients did not survive at ICU discharge and in total 5 died at hospital discharge”.

In particular, the discussion is of bad style in language terms and lengthiness. It has to be shortened. The conclusion must be concise.

There should not be repeats such as being the first to perform this study.

There are inconsistencies in the description of the parameter measurement.

 coagulopathy/anti-platelets therapy: This is not a well-chosen technical term. Authors have to describe what they mean.

In line 124 the Astute 140 meter is mentioned in contrast at line 129 the Vitros Platform is described. Which one has been used and how do these two methods compare? There are other inconsistencies such as in lines 35 and 155 compared to table 1.

There are typing errors even in the tables

Table 7 might go as supplement

Authors seem to have invested much energy in data analysis, however, if they will not be prepared to improve the quality of writing this manuscript cannot be published in a journal using English.

The quality of the presentation can only be evaluated after correction of the English wording.

Author Response

Dear Reviewer 

We have carefully reviewed the comments and have revised the manuscript accordingly. Our responses are given in a point-by-point manner below. Changes to the manuscript are shown  as track changes. We hope the revised Version is now suitable for publication and look forward to hearing from you in due course.

Thank you for your review of our paper. We have answered each of your points below:

  1. Comments and Suggestions for Authors. This study evaluates the nephrocheck parameters in trauma patients. Unfortunately, the English writing is of poor quality. There are sentences when the reader needs to guess what the authors mean. One example is this: “In our study, two of AKI patients did not survive at ICU discharge and in total 5 died at hospital discharge”.

Reply: We thanks the reviewer for this kind comment. The manuscript has now been reviewed in both scientific and English form by a mother language collegue from Australia that was added as co-author in the manuscript

  1. In particular, the discussion is of bad style in language terms and lengthiness. It has to be shortened. The conclusion must be concise.There should not be repeats such as being the first to perform this study.

Reply: We thanks the reviewer for this kind comment. The discussion was reassesed and shortened in a good language style The conclusion has been shortened.

  1. There are inconsistencies in the description of the parameter measurement. coagulopathy/anti-platelets therapy : This is not a well-chosen technical term. Authors have to describe what they mean.

Reply: We thanks the reviewer for this kind comment.  The parameter coagulopathy/anti-platelets therapy was replaced with Anti-platelet and Anti-coagulation Therapy

  1. In line 124 the Astute 140 meter is mentioned in contrast at line 129 the Vitros Platform is described. Which one has been used and how do these two methods compare? There are other inconsistencies such as in lines 35 and 155 compared to table 1.

Reply: We thanks the reviewer for this kind comment. We correct the mistake concerning vitros platform and we replaced it with astute 140 meter that was the method used. In addition, we fix the inconsistencies reported by the reviewer.

  1. There are typing errors even in the tables

Reply: We thanks the reviewer for this kind comment. We correct typing errors in the tables as reviewer’s indicate

  1. Table 7 might go as supplement

Reply: We thanks the reviewer for this kind comment. We put table7 as table supplement  errors in the tables as reviewer’s indicate

  1. Authors seem to have invested much energy in data analysis, however, if they will not be prepared to improve the quality of writing this manuscript cannot be published in a journal using English. The quality of the presentation can only be evaluated after correction of the English wording.

Reply: We thanks the reviewer for this kind comment. Hope that the manuscript after reviewer’s feedback much improved.

Reviewer 2 Report

This article by Gianlorenzo GOLINO et al. reported that the multiplicative values of TIMP2 and IGFBP7 were higher in patients with posttraumatic AKI than in non-AKI cases.
We have several concerns about this study.
1. the authors explain in the background that the purpose of this study was to identify risk factors for the development of posttraumatic AKI and to evaluate the predictive performance of the NephroCheck test compared to serum creatinine levels.
They should report on the relationship between TIMP2 and IGFBP7 and posttraumatic AKI. Still, it is unclear what they want to report, as many reports are not directly related to the biomarkers. The inclusion of these factors has changed the purpose of the study and the actual content of the analysis significantly. The authors must summarize the secondary endpoints as necessary and minimal.
2. In the statistical analysis, the authors seem to have used a stepwise backward approach to adjust for confounding factors. Still, it is unclear how the variables were actually selected and adjusted. Please add the detailed analysis procedure in the methods.
3. In the Patients Characteristics section of the Results, the authors reported that the groups that developed AKI were overweight and obese, but this may be a mistake in age.
4. The inclusion of "Renal Outcomes" in the Results section obscures the intent of this study. As mentioned above, this is not the original purpose of this study. It should be published as a separate study reporting traumatic AKI outcomes or briefly noted in the Patient Characteristics section.
5. the authors are not sure why they chose to do so, but a p-value <0.05 on univariate analysis is considered a statistically significant difference, while a positive NephroCheck score of p-value =0.055 on multivariate analysis is not considered a significant association. Also, the authors used their own TIMP2 x IGFBP7 instrument as a new diagnostic tool, not the NephroCheck, and should have data as numbers, not positives and negatives. These points are not sufficiently explained in the METHODS, and the interpretation in the RESULTS is very unclear.

Author Response

Dear Reviewer 

We have carefully reviewed the comments and have revised the manuscript accordingly. Our responses are given in a point-by-point manner below. Changes to the manuscript are shown  as track changes. We hope the revised Version is now suitable for publication and look forward to hearing from you in due course.

Thank you for your review of our paper. We have answered each of your points below:

  1. This article by Gianlorenzo GOLINO et al. reported that the multiplicative values of TIMP2 and IGFBP7 were higher in patients with posttraumatic AKI than in non-AKI We have several concerns about this study. The authors explain in the background that the purpose of this study was to identify risk factors for the development of posttraumatic AKI and to evaluate the predictive performance of the NephroCheck test compared to serum creatinine levels. They should report on the relationship between TIMP2 and IGFBP7 and posttraumatic AKI. Still, it is unclear what they want to report, as many reports are not directly related to the biomarkers. The inclusion of these factors has changed the purpose of the study and the actual content of the analysis significantly. The authors must summarize the secondary endpoints as necessary and minimal.

Reply: We thanks the reviewer for this kind comment. We underetand that it is not well explained. We better explained the aim: The present study was focused on the incidence on AKI in post-traumatic patients and also to examine risk factors associated with the development of AKI and to evalutate the combination of urinary TIMP-2 and IGFBP7 at admission to the ICU after traumatic injury identifies patients at risk for developing AKI. We strongly believe that secondary outcome should be summarized minimal but the use of other two biomarkers (TIMP-2 and   IGFBP7) in addition to serum creatinine as a good clinical result as potential strategy and tool to prevent or mitigate AKI if sufficient warning is provided, and where intervention can skew outcomes ( this part is also specified in the discussion   section).

  1. In the statistical analysis, the authors seem to have used a stepwise backward approach to adjust for confounding factors. Still, it is unclear how the variables were actually selected and adjusted. Please add the detailed analysis procedure in the methods.

We thank the reviewer for this comment. We now better explain the process of variable selection in the methods, were we now write

Multivariable logistic regression was conducted, including two clinically important trauma variables by default (ISS and shock index), and selecting other variables using stepwise backward approach according to the lowest Aikake Information Criterion (AIC). AIC depends on the number of independent variables included in the model and on the maximum likelihood estimation of each model. The best model chosen according to AIC is the one explaining the greatest amount of variation using the fewest possible independent variable.  

  1. In the Patients Characteristics section of the Results, the authors reported that the groups that developed AKI were overweight and obese, but this may be a mistake in age.

Reply: We thanks the reviewer for this kind comment. This is a mistake also noted from the first reviewer and we made a correction as you both reviwers indicated

  1. The inclusion of "Renal Outcomes" in the Results section obscures the intent of this study. As mentioned above, this is not the original purpose of this study. It should be published as a separate study reporting traumatic AKI outcomes or briefly noted in the Patient Characteristics section.

Reply: We thanks the reviewer for this kind comment. However, it is reported describing all variables involved in the renal damage

The authors are not sure why they chose to do so, but a p-value <0.05 on univariate analysis is considered a statistically significant difference, while a positive NephroCheck score of p-value =0.055 on multivariate analysis is not considered a significant association.

Reply: We thank the reviewer for this insightful comment, which allows us to better explain in the methods that the variables were included in the logistic regression model according to Aikake Information Criterion. Nephrocheck score at admission was significantly associated with mortality at univariate analysis, while maintaining a borderline significance (p=0.055) at multivariable analysis as reported in the results, still contributing to overall model information (AIC and borderline wald test). We now have better explained this aspect in methods, and in article conclusion.

  1. Also, the authors used their own TIMP2 x IGFBP7 instrument as a new diagnostic tool, not the NephroCheck, and should have data as numbers, not positives and negatives. These points are not sufficiently explained in the METHODS, and the interpretation in the RESULTS is very unclear.

Reply: We are grateful to the reviewer for point out this possible problem in our methods. We actually used NephroCheck and not a proprietary TIMP2 x IGFBP7, thus results are according to AkiRisk Nephrocheck scores. We now write in methods The final test output, labeled AKI Risk, is shown as a numeric score, and is considered as continuous variable in this article.

Reviewer 3 Report

This is a study evaluating the utility of nephrocheck in the diagnosis of AKI related outcomes in a trauma ICU setting. This is of interest to the field and is an interesting study. I have a few questions and comments :

  1. In Table 1 the age is significantly different in the two groups suggesting that the AKI group was much older. How have the authors accounted for this? Was this adjusted for in their regression analysis. Please list beta coefficients for each variable.
  2. The authors show Cr trend in AKI vs no AKI - however the diagnostic criteria per their definition uses Cr as a marker /criteria. Hence this figure maybe redundant and needs to be clarified
  3. How have the authors defined AKI in Figure 3? If Cr is a criteria then why doesnt it correlate better with AKI?If they used a more sensitive criteria like cystatin C then this should be stated and the figure statistically would be appropriate.
  4. How did the authors define regression and progression of AKI?
  5. Did the authors evaluate longitudinal nephrocheck values in individuals with AKI? For those needing RRT , did this show a change during progression of AKI?
  6. Nephrocheck seems to correlate with hypotension? In the hypotension group and in the non hypotension group - please plot who got AKI vs NO AKI?
  7. Like table 2, please consider adding an additional table with the same characteristics but divided in to groups per nephrocheck levels. This would be more illuminating. 
  8.  

Author Response

Dear Reviewer 

We have carefully reviewed the comments and have revised the manuscript accordingly. Our responses are given in a point-by-point manner below. Changes to the manuscript are shown  as track changes. We hope the revised Version is now suitable for publication and look forward to hearing from you in due course.

Thank you for your review of our paper. We have answered each of your points below:

  1. This is a study evaluating the utility of nephrocheck in the diagnosis of AKI related outcomes in a trauma ICU setting. This is of interest to the field and is an interesting study. I have a few questions and comments.

Reply:  We thanks the reviewer for this kind comment. We really appreciate your compliment.

  1. In Table 1 the age is significantly different in the two groups suggesting that the AKI group was much older. How have the authors accounted for this? Was this adjusted for in their regression analysis. Please list beta coefficients for each variable.

Reply: We thank the reviewer for this comment. We accounted for this in logistic regression model, were age was included as a variable, and remained significant after controlling for other variables. Odds ratios for multivariable regression were reported in the supplemental material. We list here below beta coefficients for each variable as requested.

Characteristic

OR1

95% CI1

p-value

Age

0.04 (0.02; 0.07)

1.04

1.02, 1.07

0.002

Male

1.0 (-0.22; 2.5)

2.82

0.80, 12.3

0.13

Shock Index

0.87 (-0.56; 2.2)

2.40

0.57, 9.41

0.2

ISS

-0.03 (-0.08; 0.02)

0.97

0.92, 1.02

0.3

Previous cardiac disease

1.3 (0.05; 2.7)

3.75

1.05, 15.0

0.048

[TIMP-2] × [IGFBP7]

Admission*

0.43 (-0.01; 0.87)

1.53

0.99, 2.40

0.055

  • The authors show Cr trend in AKI vs no AKI - however the diagnostic criteria per their definition uses Cr as a marker /criteria. Hence this figure maybe redundant and needs to be clarified

Reply:  We thanks the reviewer for this kind comment. We decide to remove this figure because as reviewer’s indicate it is redundant

  1. How have the authors defined AKI in Figure 3? If Cr is a criteria then why doesnt it correlate better with AKI?If they used a more sensitive criteria like cystatin C then this should be stated and the figure statistically would be appropriate.

Reply: we thank the reviewer for the comment. Association between Nephrocheck (A) and Serum Creatinine at admission (B) and  incidence of AKI at 24 hours. AKI was assessed based on KDIGO Criteria. We did not used Cystatin C.

  1. How did the authors define regression and progression of AKI?

Reply: Thanks a lot for you comment. Progression of AKI was defined as  increase in sCr ≥1.5 times baseline, which is known or presumed to have occurred within the prior 7 days; while regression was defined a decrease sCr ≥1.5 times baseline with 7 days. We added in the manuscript in the definitions section

  1. Did the authors evaluate longitudinal nephrocheck values in individuals with AKI? For those needing RRT , did this show a change during progression of AKI?

Reply: We thank the reviewer for this comment. The number of patients needing RRT was too low for allowing any inference on the nephrocheck values over time, thus we could not evaluate this aspect

  1. Nephrocheck seems to correlate with hypotension? In the hypotension group and in the non hypotension group - please plot who got AKI vs NO AKI?

We thank the reviewer for this comment, we report here below the boxplot of nephrocheck values according to hypotension episodes

  1. Like table 2, please consider adding an additional table with the same characteristics but divided in to groups per nephrocheck levels. This would be more illuminating.

We thank the reviewer for this comment, which helps us further clarify our data.   We now add in supplemental material an additional table as requested, reporting renal outcomes according to nephrocheck values

Supplemental table 2: Table 2. Renal Outcomes according to [TIMP-2] × [IGFBP7] values at admission  

Characteristic

Entire Cohort= 1531

[TIMP-2] × [IGFBP7] Admission <=0.03, N = 501

[TIMP-2] × [IGFBP7] Admission > 0.3 AKI, N = 821

p-value2

sCr Admission

0.84 (0.70, 1.01)

0.82 (0.70, 0.99)

0.87 (0.71, 1.08)

0.3

sCr 24hrs

0.82 (0.70, 0.99)

0.80 (0.72, 0.95)

0.86 (0.70, 1.05)

0.2

sCr 48hrs

0.81 (0.68, 1.07)

0.78 (0.68, 0.92)

0.84 (0.72, 1.09)

0.3

sCr 72hrs

0.75 (0.64, 0.92)

0.74 (0.64, 0.85)

0.74 (0.66, 0.92)

0.8

Myoglobin Admission

974 (458, 1,888)

964 (411, 2,888)

979 (518, 1,664)

0.8

Myoglobin 24hrs

1,354 (630, 2,705)

1,449 (662, 2,856)

1,362 (643, 2,741)

0.8

Myoglobin 48hrs

697 (258, 1,595)

774 (372, 1,076)

710 (211, 2,728)

0.7

Myoglobin 72hrs

835 (442, 2,236)

665 (442, 1,680)

893 (371, 1,366)

>0.9

Glucose Admission

150 (128, 179)

158 (131, 189)

148 (128, 177)

0.4

Glucose 24hrs

135 (118, 160)

143 (115, 167)

135 (121, 161)

0.7

Glucose 48hrs

129 (114, 152)

130 (114, 149)

129 (118, 153)

0.8

Glucose 72hrs

129 (113, 150)

137 (114, 156)

127 (116, 146)

0.4

PCT Admission

0.27 (0.07, 1.15)

0.14 (0.06, 0.54)

0.34 (0.11, 0.94)

0.5

PCT 24hrs

0.87 (0.32, 3.05)

1.65 (0.14, 5.10)

0.62 (0.33, 2.81)

0.5

PCT 48hrs

0.9 (0.5, 3.9)

1.3 (0.8, 3.9)

1.6 (0.3, 4.6)

0.7

PCT 72hrs

0.7 (0.3, 2.1)

1.3 (0.8, 2.8)

0.4 (0.3, 2.1)

0.063

FB 24hrs

1,430 (457, 2,636)

1,441 (498, 3,100)

1,307 (446, 2,620)

0.8

FB 48hrs

850 (-16, 1,938)

727 (110, 1,823)

1,012 (61, 1,946)

0.7

FB 72hrs

860 (102, 1,671)

555 (104, 1,263)

1,021 (289, 2,004)

0.2

% FO 24hrs

18 (5, 35)

18 (8, 36)

16 (5, 35)

0.9

% FO 48hrs

11 (0, 25)

11 (1, 25)

13 (1, 25)

0.7

% FO 72hrs

11 (1, 20)

7 (1, 17)

14 (4, 24)

0.3

UO 24hrs

1,855 (1,289, 2,575)

1,910 (1,340, 2,950)

1,942 (1,322, 2,555)

0.7

UO 48hrs

2,300 (1,725, 3,000)

2,350 (1,725, 3,410)

2,110 (1,650, 2,850)

0.2

UO 72hrs

2,570 (1,888, 3,140)

2,775 (2,378, 3,442)

2,190 (1,770, 2,960)

0.015

Furosemide 24hrs

42 (28%)

12 (24%)

27 (33%)

0.3

Furosemide 48hrs

41 (36%)

16 (41%)

22 (37%)

0.7

Furosemide 72hrs

22 (26%)

9 (35%)

12 (24%)

0.4

RRT 24hrs

1 (0.7%)

0 (0%)

0 (0%)

>0.9

RRT 48hrs

4 (3.5%)

0 (0%)

3 (5.0%)

0.3

RRT 72hrs

4 (4.7%)

0 (0%)

3 (6.1%)

0.5

1Median (IQR); n (%). x2Wilcoxon rank sum test; Wilcoxon rank sum exact test. ACRONYM: FB, Fluid Balance; FO, Fluid Overload; PCT, procalcitonine; sCr, serum creatinine;  RRT, renal replacement therapy;  UO, urinary output;.

Round 2

Reviewer 1 Report

this manuscript is now much improved in its presentation. There is a problem with the wording on lines 362-363. It has to be clarified before publication. 

The figure on page #8 has either to be taken out as the data are shown in one of the tables or it needs a figure legend.